# Multiomics Analysis of the Mechanism by Which Gibberellin Alleviates S-Metolachlor Toxicity in Rice Seedlings

**DOI:** 10.3390/plants13172517

**Published:** 2024-09-07

**Authors:** Cong Wang, Haona Yang, Zhixuan Liu, Lianyang Bai, Lifeng Wang, Shangfeng Zhou

**Affiliations:** 1Longping Branch, Graduate School of Hunan University, Changsha 410125, China; irisiiii@hnu.edu.cn; 2Hunan Agricultural Biotechnology Research Institute, Hunan Academy of Agricultural Sciences, Changsha 410125, China; haonayang@hunaas.cn; 3Hunan Rice Research Institute, Hunan Academy of Agricultural Sciences, Changsha 410125, China; zhixuanliu@hunaas.cn

**Keywords:** herbicide, transcriptome, proteomics, metabolomics

## Abstract

S-metolachlor is a selective pre-emergence herbicide used in dryland. However, it is challenging to employ in paddy fields due to its phytotoxic effects on rice. As a common phytohormone, Gibberellin-3 (GA_3_) is inferred to have the ability to alleviate herbicide phytotoxicity. This study first quantitatively verified the phytotoxicity of s-metolachlor to rice and then demonstrated the mitigative effect of GA_3_ on these adverse reactions. Furthermore, a transcriptome of rice seedlings subjected to different treatments was constructed to assemble the reference genes, followed by comparative metabolomics and proteomics analyses. Metabolomics revealed an enrichment of flavonoid metabolites in the group of adding GA_3_, and these flavonoids can eliminate ROS in plants. Proteomics analysis indicated that differential proteins were enriched in the phenylpropanoid biosynthesis pathway responsible for the synthesis of flavonoids and that the functions of most differential proteins are associated with peroxidase. The proteome, combined with the transcriptome, revealed that the expressions of proteins and genes was related to the POD activity in the group of adding GA_3_. It was speculated that the elimination of ROS is key to alleviating the stress of s-metolachlor on rice growth. It was inferred that the mechanism of GA_3_ in alleviating the phytotoxicity of the substance s-metolachlor is by increasing the activity of the POD and influencing the growth of rice seedlings through the restoration of flavonoid synthesis. In this study, we screened GA_3_ as a safener to alleviate the phytotoxicity of s-metolachlor on rice. On this basis, the mechanism of alleviating phytotoxicity was studied. The application range of s-metolachlor might be expanded, providing a new supplementary method for weed control and herbicide resistance management.

## 1. Introduction

Rice (*Oryza sativa* L.) is a major food crop that feeds half of the world’s population [1]. Weeds are one of the most critical factors influencing rice yield reduction [2]; various chemical, ecological, and physical methods have been employed to mitigate their negative effects in paddy fields. As is well known, herbicides are the most effective method for controlling harmful agricultural weeds due to the advantages of time saving, simplicity, and high efficiency. In the last few decades, herbicides have largely recovered yield losses caused by weeds and have become one of the main methods of weed control [3].

S-metolachlor is the main pre-emergence herbicide used to control weeds in dryland, primarily targeting annual monocotyledons and sedges due to its high efficiency and broad-spectrum coverage. The herbicidal activity of the herbicide is 1.4 to 1.6 times that of glyphosate, with greater selectivity and a shorter half-life [4]. The mechanism of action of s-metolachlor is to inhibit protease activity [5]. S-metolachlor belongs to the class of amide herbicides, inhibiting lipid synthesis, cell division, and growth. S-metolachlor inhibits photosynthesis in plants. [6]. The application of S-metolachlor has been demonstrated to disrupt the permeability of cell membranes in the roots of both maize and rice [7]. Liu clearly stated that 0.5 mg/L s-metolachlor can reduce the content of chlorophyll a and chlorophyll b in algae by 92.9% and 91.1%, respectively, leading to an imbalance of plant hormones [8]. However, rice is sensitive to s-metolachlor, which can affect the growth of rice after application, leading to a reduction in rice yield. Despite this, s-metolachlor is still one of the widely used herbicides for weed control in paddy fields.

Given that s-metolachlor can induce phytotoxicity in crops, it is essential to alleviate the herbicide’s phytotoxicity. Safety agents are a type of compound that can protect crops and reduce herbicide damage without affecting herbicide activity [9]. Safety agents play an important role in expanding the application range of herbicides. Various safety agents are used to alleviate the harm caused by glyphosate. Commercial safety agents commonly used to alleviate the harm caused by glyphosate include AD-67 and chlorpromazine [10]. Exogenous chemical regulators, such as herbicide safeners, can also induce stress tolerance in plants [11]. The germination of rice seeds, the expansion of leaves, the onset of flowering, and the maturation of pollen are processes that are associated with gibberellins (GA). In particular, the transition from nutrient growth to flowering, and the subsequent elongation of the stem, are influenced by GA [12,13]. Plants regulate the various functions of GA by coordinating the activation and inactivation of multiple biosynthetic genes in different tissues at various developmental stages [14]. Exogenously applied GA has been shown to enhance the antioxidant capability of *Cerasus humilis* [15]. GA_1_, GA_3_, GA_4_, and GA_7_ are biologically active in higher plants. Among them, GA_3_ is the most active and widely distributed gibberellin, having been widely used as a hormone safener in alleviating herbicide toxicity [16]. However, the molecular mechanism of GA detoxification remains unclear.

The aim of this study was to explore the relationship between GA_3_ and s-metolachlor toxicity and to verify whether GA_3_ can alleviate the toxicity of herbicides to rice and their detoxification mechanisms. This study applied GA to investigate the alleviation of phytotoxicity caused by the herbicide s-metolachlor. Subsequently, changes in metabolite synthesis were analyzed using metabolomic analysis, while transcriptomic and proteomic analyses were conducted to elucidate the detoxification mechanism in general. This paper demonstrates that gibberellins can be detoxified by scavenging reactive oxygen species in plants, thus providing a safe agent that allows the dryland herbicide s-metolachlor to be used as a pre-emergence herbicide for the safe control of weeds in rice fields.

## 2. Results

### 2.1. GA Recovered S-Metolachlor Inhibition of Rice Seedlings

To verify the alleviating effects of GA_3_ on rice seedlings injured by s-metolachlor, three physiological indicators were monitored, including shoot length and root length. Averages shoot lengths were 132.71, 81.96, 119.54, and 132.21 mm in the control (CK), addition of s-metolachlor (M2), addition of s-metolachlor and GA_3_ (M3), and addition of GA_3_ (M4) groups, respectively. Average root lengths were 83.71, 48.29, 58.63, and 59.25 mm, respectively, in the CK, M2, M3, and M4 treatments (Figure 1B). Compared with CK, the inhibition rates of shoot length and root length were 38.24% and 42.31% in M2, respectively; recovered to 9.92% and 29.9% in M3, respectively; and remained at in 14.99%, 29.22% in M4, respectively. The addition of exogenous GA_3_ significantly alleviated the phytotoxicity (Figure 1A), and the mitigation effect was mainly observed in the stem, leaf, and root parts of the crop.

### 2.2. S-Metolachlor and GA_3_ Affect the Activity of Various Enzymes

The contents of H_2_O_2_ and the activities of three antioxidant oxidases including peroxidase (POD), superoxidase dismutase (SOD), and glutathione S-transferase (GST) were analyzed to investigate the physiological and biochemical alterations in rice seedlings exposed to treatments of s-metolachlor and GA_3_. The contents of H_2_O_2_ were 2.02, 3.33, 1.65, and 1.59 mmol/gprot, respectively, in CK, M2, M3, and M4 (Figure 2A). The activities of POD were 40.08, 32.98, 74.73, and 76.88 (U/mg protein) (U/mg protein is the unit of the enzyme activity, which represents the activity of the enzyme in mg of protein), respectively, in CK, M2, M3, and M4 groups; this index in M2 was 17.71% lower than CK, and that in M3 was 86.45% higher than CK (Figure 2B); The activities of SOD were 10.69, 9.39, 19.85, and 19.32 (U/mg protein), respectively, in CK, M2, M3, and M4; this index in M2 was 12.16% lower than CK, and that in M3 was 85.69% higher than CK, respectively (Figure 2C); and the activities of GST were 46.77, 35.58, 69.41, and 54.62 (U/mg protein), respectively, in CK, M2, M3, and M4; there was a 23.93% activity decrease in M2 and 48.41% activity increase in M3, respectively (Figure 2D). The H_2_O_2_ content in M2 was significantly higher than the other three groups, while POD, SOD, and GST activity in the M3 group was significantly higher than in M2.

### 2.3. GA_3_ Alleviates Toxicity by Affecting the Synthesis Pathway of Flavonoids

To dissect the mechanism of the above phenomenon, we detected all metabolites in different groups. After PCA analysis (Appendix A) and mapping in the database, we annotated 34, 52, and 44 differential metabolites, respectively, in the groups of CKvsM2, CKvsM3, and M2vsM3 (Figure 3A). The types of metabolites detected in CKvsM3 were organic acids and derivatives, alkaloids, flavonoids, and lipids (Figure 3B). In the CKvsM3 group, the main metabolites were flavonoids, alkaloids, HCAAs, and lipids (Figure 3C), In the M2vsM3 comparison, the compounds identified were HCAAs, flavonoids, alkaloids, vitamins and their derivatives, nucleotides and their derivatives, and lipids. Moreover, the majority were enriched in HCAAs and flavonoids (Figure 3D). KEGG enrichment analysis was conducted using all reference metabolites as background to understand the functions of DEGs. The bubble chart shows that the pathway with the highest accumulation of differential metabolites was the flavonoid pathway, followed by cysteine and methionine metabolism, tryptophan metabolism, and the biosynthesis of alkaloids derived from the shikimate pathway (Figure 3E). In the CKvsM4 group, the main pathways were metabolic pathways and biosynthesis of secondary metabolites (Appendix A).

### 2.4. GA_3_ Affects Gene Expression on Multiple Pathways

In order to identify the differentially expressed genes associated with alleviating phytotoxicity, high-throughput sequencing was conducted on the mRNA isolated and purified from four groups of rice samples treated with different treatments. Using a comparative analysis of DEG spectra, the differential gene expression was identified in three groups of samples treated with CK, refined metolachlor, and refined metolachlor and gibberellin. The screening conditions for differential genes were genes with |log2Fold Change| ≥ 1 and FDR < 0.05. A total of 270 DEGs were identified in the CKvsM2 group, including 195 down-regulated DEGs and 75 up-regulated DEGs; a total of 465 DEGs were identified in the CKvsM3 group, including 266 down-regulated DEGs and 199 up-regulated DEGs; a total of 687 DEGs were identified in the M2vsM3 group, including 262 down-regulated DEGs and 425 up-regulated DEGs (Figure 4A) (Appendix A). The differentially expressed genes in the M2vsM3 group may only differ in CKvsM2 but not in CKvsM3. There were 70 genes with no differences in CKvsM3 and 73 genes with only differences in CKvsM3 but no differences in CKvsM2 (Figure 4B).

To explore the function of the above DEGs in M2vsM3, the annotation of KEGG pathways was adopted. It should be noted that the selected pathways were those annotated DEGs over 10 due to the huge number of DEGs. It was found that 56 out of 165 genes were annotated in metabolic pathways and 41 out of 165 genes annotated in the biosynthesis of secondary metabolites (Figure 4C). In addition, 11 DEGs were annotated in the phenylpropanoid biosynthesis pathway, which is the upstream pathway of flavonoid biosynthesis.

### 2.5. GA_3_ Affects Multiple Protein Synthesis Pathways

Since we focused on the mitigating effect of gibberellin addition on herbicide toxicity, we subsequently selected only CKvsM2, CKvsM3, and M2vsM3 for proteomics analysis. In order to elucidate the mechanism by which GA_3_ alleviates the phytotoxicity of refined metolachlor on rice seedlings, this experiment also utilized iTRAQ technology to quantitatively analyze the rice stem and leaf samples from different groups and identify them using mass spectrometry. There were 318,489 spectra, and the number of available effective spectra was 96,759. A total of 47,331 peptides were identified through spectral analysis, of which the specific peptides were 43,416. A total of 7528 proteins were identified, of which 6467 were quantifiable. Taking the change in differential expression over 1.5 as the change threshold for significant up-regulation and less than 1/1.5 as the change threshold for significant down-regulation, the numbers of differential proteins in each group were counted. Among them, there were 208 down-regulated proteins and 111 up-regulated proteins in the CKvsM2 group; 36 down-regulated proteins and 73 up-regulated proteins in the CKvsM3 group; and 64 down-regulated proteins and 169 up-regulated proteins in the M2vsM3 group (Figure 5A) (Appendix A).

To further elaborate on the mechanism, GO analysis was performed. 343, 167, and 248 were enriched in biological process (BP), cellular process (CC), and molecular function (MF), respectively (Figure 5B). In biological process, DAPs were enriched in metabolic process, cellular process, and single-organism process. In the cellular component, meanwhile, DAPs were enriched in cell and membrane. In addition, binding and catalytic activities enriched most DAPs in molecular function. According to the cluster analysis of the KEGG pathway, the differential proteins in the proteome were mainly enriched in phenylpropanoid biosynthesis (ko00940) (Figure 5C). A network of differential proteins annotated in ko00940 was built (Figure 5D). It was found that the core protein was A2WN51, which connected with A2YIB0, A2Y667, and B8ARU3 via A2Z9A1. The protein descriptions of A2WN51, A2YIB0, A2Y667, and B8ARU3 are all about peroxidase. A2Z9A1 is a protein associated with cytochrome P450 (Table 1).

### 2.6. Conjoint Analysis

The differential proteins and metabolites in the phenylpropanoid biosynthesis pathway and flavonoid pathway were annotated. The phenylpropanoid biosynthesis pathway is the upstream pathway of the flavonoid pathway, and four proteins were annotated in this pathway: *PTAL*, *4CL*, *F5H*, and *CAD*. On the other hand, the flavonoid pathway annotated two genes: *Os04g0611400* and *Os04g0581000.* In addition, the contents of certain metabolites had changed, such as prunin, naringin, luteolin, eriodictyol, dihydromyricetin, delphinidin, naringenin, dihydrokaemperol, kaempferol, quercetin, phloretin, phlorizin, butin, chrysin, and isoliquiritigenin (Figure 6).

### 2.7. Q-PCR Validation of Candidate Genes

To verify the results of RNA-seq, 6 genes were selected from phenylpropanoid biosynthesis and flavonoid biosynthesis. The results were similar to those of RNA-seq, and the data sdemonstrated consistency in the up- and down-regulated genes within these pathways (Figure 7). Meanwhile, the relative expression of *Os04g0611400*, *Os06g0522300*, *Os07g0677500*, *Os09g0262000*, *Os10g0109600*, and *Os12g0199500* in M3 were all higher than that in M2.

## 3. Discussion

### 3.1. GA_3_ Recovered S-Metolachor Inhibition of Rice Seedlings

S-metolachlor [2-chloro-N-(2-ethyl-6-methylphenyl)-N-(methoxy-1-methylethyl) acetamide] is useful for weed management in general and has been used for more than 60 years [17,18]. Studies have shown that using s-metolachlor can cause toxicity to rice [19]. It is of great significance to broaden the application scope of this herbicide by alleviating pesticide damage in rice and enabling its use as a sealing agent for water in rice seedlings. From the measurement data of physiological indicators, it could be verified that s-metolachlor had a certain phytotoxicity to the crop itself and that GA_3_ provides a certain alleviating effect on injured rice seedlings. This is beneficial for protecting the normal growth of rice. However, the root growth of rice seedlings supplemented with only GA_3_ was inhibited. In recent research, it was reported that Gibberellin inhibits taproot formation by modulating the DELLA-NAC complex activity in turnip [20]. Therefore, the mechanism of GA_3_ inhibition on rice roots needs to be studied.

### 3.2. GA_3_ Alleviates S-Metolachlor Toxicity by Affecting Enzyme Activity

Figure 2 clearly shows that the activities of these three enzymes all decreased with the addition of s-metolachlor (M2) compared with the CK groups and that the reduction of SOD and POD reached a significant level. However, the activity of SOD and POD were significantly increased, while the contents of H_2_O_2_ were reduced by GA_3_ supplement (M3 and M4), with or without the addition of s-metolachlor. H_2_O_2_ is an active oxygen species produced during metabolism; SOD and POD are important oxidases for animals and plants. SOD scavenges excessive free radicals and plays an important role in the plant’s self-protection system. In addition, POD resists and scavenges ROS, inhibits and prevents cell membrane peroxidation, and maintains membrane system stability in rice [21]. The supplementation of exogenous GA_3_ could enhance the activity of SOD and POD in rice seedlings even in the absence of s-metolachlor stress, helping to reduce external stress. Additionally, the well-recovered rice seedlings exhibited high levels of SOD and POD activity. In addition, compared to the M2 group, protein A associated with POD activity was up-regulated in the M3 group (Table 1). Adding GA_3_ was found to enhance the activity of POD enzyme, eliminate ROS, and alleviate the phytotoxicity of s-metolachlor on rice seedlings.

The variation trend of GST activity is more complex. As can be seen from Figure 3C, there was no significant variation in GST activity with the individual addition of s-metolachlor (M2) or GA_3_ (M4) compared with the CK group. However, a significant increase in GST activity was observed in the treatment group M3, which was treated with both s-metolachlor and GA_3_. That was consistent with the work of Wu et al., who found that the GST activity in crops treated with metformin was lower than that in the control group and GST activity increased after the addition of an alleviating compound [22]. In plants, the key step in the detoxification of herbicides involves the interaction between catalytically oxidative cytochrome P450 monooxygenase (CYP) and electrophilic herbicides with the tripeptide glutathione (GSH). This reaction is catalyzed by glutathione transferase (GST) to achieve detoxification [23,24]. The mechanism of action of s-metolachlor is to inhibit protease activity [5]. The results above indicated that exogenous GA_3_ has a specifically enhanced GST activity in rice seedlings and that the phytotoxicity of s-metolachlor might invoke this detoxification response, and the detailed responses of GST to s-metolachlor damage are worth studying in depth.

### 3.3. Adding GA_3_ Affects the Synthesis of Flavonoids

Metabolites, as crucial products in the life activities of organisms, represent manifestations of biological phenotypes; they are conducive to effectively researching various biological phenomena [25]. Plant metabolites are mainly categorized into two categories: primary metabolites and secondary metabolites. They maintain the life activities, growth and development, and participate in the response to biological and abiotic stress of plants [26]. Flavonoids mainly include flavones, flavonol, flavanones, flavanonol, anthocyanidins, flavan-3,4-diols, xanthones, chalcones, biflavonoids, and so on. The antioxidant properties of flavonoids stem from their ability to scavenge and neutralize ROS, stabilising free radicals by donating electrons or hydrogen atoms [27]. Studies have shown that the protective effect of flavonoids stems from their ability to inhibit lipid peroxidation, chelate redox-active metals, and weaken other processes involving ROS [28]. Phenolamines are also often referred to as hydroxycinnamic acid amides (HCAAs) [29]. Phenolamine is a secondarymetabolite that plays a crucial role in the growth and development of plants, helping them resist stress and adversity. The resistance of phenamine to adversity appears to primarily rely on its antioxidant activity and its ability to scavenge free radicals. Phenolamines are suitable substrates for peroxidase, and they are also effective in removing hydrogen peroxide [30]. The enzyme activity analysis revealed that the POD activity in rice seedlings treated simultaneously with s-metolachlor and GA_3_ increased, suggesting that the recovery of the phenotype may be associated with the elimination of ROS in the rice seedlings. Therefore, we chose the flavonoid biosythesis pathway as a candidate research pathway in proteome and transcriptome research.

As the upstream pathway of the flavonoid pathway, the phenylpropanoid pathway is one of the important production pathways of secondary metabolites in plants [31]. A variety of enzymes are involved in this metabolic pathway, among which phenylalanine ammonia lyase (*PAL*), cinnamate-4-hydroxylase (*C4H*), and 4-coumarate-CoA ligase (*4CL*) are important [32]. *PTAL* (phenylalanine/tyrosine ammonia-lyase) belongs to the aromatic amino acid lyase family, which also includes histidine ammonia-lyase (*EC 4.3.1.3*), tyrosine ammonia-lyase (*EC 4.3.1.23*), and phenylalanine ammonia-lyase (*EC 4.3.1.24*). The enzyme from some monocots deaminates L-phenylalanine and L-tyrosine with similar catalytic efficiency [33]. *4CL* plays an important role in the channelized flux of synthesizing different phenylpropane derivatives in plants [34]. It is a key branch point enzyme that converts cinnamic acid derivatives into coenzyme A thioesters. These intermediates can be converted into phenylpropanoid-derived compounds, such as phenolic glycosides, phenylpropanoid esters, and lignin, as well as flavonoids [32]. Flavanone 3-hydroxy-lase (*F3H*) is a key enzyme at the branching point of the flavonoid synthesis pathway, which catalyzes the hydroxylation of 4,5,7-flavanone C3 to generate dihydrogen Kaempferol (dihydrok-aempferol, DHK), and DHK serves as the precursor for the biosynthesis of flavonols and other flavonoids [35,36]. In this study, we found that the expression levels of PTAL and 4CL in the M3 group were upregulated compared to the M2 group but close to the CK group. The expression level of F3H gene in the M3 group was upregulated. This indicated that, compared with only adding s-metolachlor, adding GA_3_ at the same time can restore the vitality of *PTAL* and *4CL*, which are crucial enzymes for the subsequent synthesis of flavonoids. Therefore, we speculate that the vitality of the upstream affects the production of flavonoids in the downstream, thereby influencing the growth of rice seedlings.

## 4. Materials and Methods

### 4.1. Plant Materials and Treatments

Dry mature rice (Xiang Zao Xian 45#) seeds were germinated under light conditions at 25 °C. Germination time was 2d. The culture medium was prepared using agar, sterile water, and s-metolachlor. The germinated rice seeds were planted in the agar, with eight plants per group and three replicates for each treatment.

S-metolachlor (960 g/L, EC) was purchased from Swiss Syngenta Crop Protection Co., Ltd. (Shanghai, China). GA was purchased from Macklin Company (Shanghai, China). Agar was supplied by Beijing Hongrun Baoshun Technology Co. Ltd. (Beijing, China). Previous studies have shown that 0.25 µM s-metolachlor has inhibitory effects on rice growth. Therefore, the germination medium for three experimental groups [37], including refined metolachlor (M2), refined metolachlor and GA (M3), or GA alone (M4), and one control group (CK) consisted of 5L 0.3% agar medium. In addition, M2 contained 0.25 µM s-metolachlor, M3 contained 0.25 µM s-metolachlor and 8 mg/L GA_3_, and M4 contained 8 mg/L GA_3_. After implantation, groups of rice seeds were cultivated for 7 days in a climate chamber under light (14 h at 30 °C) and dark (10 h at 27 °C) conditions, and the relative humidity was controlled at 60%. Plant height, root length, and fresh weight were measured 7d after treatment. The subsequent samples used are all powdered after freezing with liquid nitrogen and grinding.

### 4.2. Determination of Enzyme Activities

Samples were ground and homogenized with liquid nitrogen five days after treatment. Total protein levels of superoxide dismutase (SOD), peroxidase (POD), and glutathione thiol transferase (GST) were determined spectrophotometrically using commercial assay kits purchased from Nanjing Jiancheng Institute of Bioengineering. Total proteins were quantified using the Bradford assay [38], and glutathione S-transferase (GSTs) was detected using a colorimetric method [39] based on the principle that the oxidation of glutathione (GSH) and hydrogen peroxide (H_2_O_2_) can be catalyzed by GSH-Px to produce oxidized glutathione (GSSG) and H_2_O. The total superoxide dismutase (T-SOD) content was assayed using the xanthine oxidase (WST-1) method [40] based on the production of O_2_^−^ anions. Peroxidase (POD) activity was measured by monitoring the change in absorbance at 420 nm resulting from the catalysis of H_2_O_2_. H_2_O_2_ is catalyzed by a 405 nm optical path.

### 4.3. Metabolome Analysis

The freeze-dried leaf was pulverized using a mixer mill (MM 400, Retsch) with a zirconia bead for a period of 1.5 min at a frequency of 30 oscillations per minute. One hundred milligrams of powder was weighed and extracted overnight at 4 °C with 1.0 ml of 70% aqueous methanol. Subsequently, the extracts were subjected to centrifugation at 10,000 rpm for 10 min. Following this, the extracts were absorbed using a CNWBOND Carbon-GCB SPE Cartridge (250 mg, 3 ml; ANPEL, Shanghai, China; http://www.anpel.com.cn/, accessed on 1 December 2023) and filtered using a SCAA-104 filter (0.22 μm pore size; ANPEL, Shanghai, China; http://www.anpel.com.cn/) prior to LC-MS analysis. The extract was then filtered (SCAA-104, 0.22 μm pore size; ANPEL, Shanghai, China, http://www.anpel.com.cn/) and prepared for LC-MS analysis. The standard is as follows: The solution should be prepared by dissolving the substance in dimethyl sulfoxide (DMSO) or methanol and stored at −20 °C. Prior to mass spectrometry analysis, the solution should be diluted with 70% methanol to an appropriate concentration. The solvents methanol, acetonitrile, and ethanol were obtained from Merck (Darmstadt, Germany), while the standard samples were sourced from BioBioPha(Kunming, China) (http://www.biobiopha.com/, accessed on 1 December 2023) and Sigma Aldrich (http://www.sigmaaldrich.com/united-states.html, accessed on 1 December 2023). Twelve samples were divided into four groups and used for metabolomics analysis. Ionization modes include positive ion mode and negative ion mode. Samples were analyzed by ultra-performance liquid chromatography (UPLC) (Shim-pack UFLC SHIMADZU CBM30A, https://www.shimadzu.com.cn/, accessed on 1 December 2023) and tandem mass spectrometry (MS/MS) (Applied Biosystems 6500 QTRAP, https://www.thermofisher.cn/cn/zh/home/brands/applied-biosystems.html, accessed on 1 December 2023). We quantified the metabolites and provided secondary annotations. The functional annotation and analysis of pathways were performed using the Kyoto Encyclopedia of Genes and Genomes (KEGG) (https://www.genome.jp/kegg/) (accessed on 1 December 2023). We conducted principal component analysis (PCA) and cluster analysis on metabolites (www.r-project.org/, accessed on 1 December 2023) and compared the metabolites present in the different groups of rice to identify changes in the quantity of metabolites between treatments (log2Fold change ≥ 1). METWARE company completed (Wuhan, China).

### 4.4. RNA-Seq Analysis

Twelve samples collected from four groups were used for RNA-seq analysis. Total RNA was extracted using Trizol^®^ (Vazyme, Nanjing, China). Eukaryotic mRNA was enriched by binding Oligo (dT) magnetic beads to the ployA tail of mRNA, and then the mRNA was randomly interrupted. Using fragmented mRNA as a template and 6-base random primers, the first strand of cDNA was synthesized in the M-MuLV reverse transcriptase system. Subsequently, RNA chains were degraded using RNaseH (Thermofisher, Waltham, MA, USA), and the second strand of cDNA was synthesized using dNTPs as raw materials in the DNA polymerase I system. The purified double-stranded cDNA was subjected to end repair, followed by the addition of an A tail and ligation with sequencing adapters. cDNA of about 200 bp was screened using AMPure XP beads (Beckman coulter, Brea, CA, USA), PCR amplification was performed, and the PCR product was purified again using AMPure XP beads to obtain a library. After the library construction was completed, it was quantitateed using a Qubit 2.0 fluorescence meter (Thermofisher, Waltham, MA, USA). After diluting the library to 1.5 ng/μL, the inserted fragments of the library were detected using an Agilent 2100 Bioanalyzer (Agilent, City of Santa Clara, CA, USA). If the length of the inserted fragment meets expectations, then use quantitative real-time PCR (qRT-PCR) to accurately quantify the effective concentration of the library (effective concentration of the library is higher than 2 nM) After passing the inventory inspection, sequencing was performed using the Illumina HiSeq platform (Illumina, San Diego, CA, USA). Twelve samples collected from four groups were used for RNA-seq analysis. Total RNA was extracted using Trizol^®^, and the total number of differentially expressed genes, the number of up-regulated genes, and the number of down-regulated genes were calculated after the analysis of the differentially expressed genes using DESeq2. METWARE company completed (Wuhan, China).

### 4.5. Proteomics Analysis

The stems and leaves of each treatment group were taken, and then liquid nitrogen was added and ground to powder. Next, 4 times the volume of phenol extraction buffer was added, followed by Tris-balanced phenol after ultrasonic lysis and centrifugation. After the supernatant had been collected, methanol and acetone were added for washing, followed by re-dissolving the precipitate with urea to finalize the protein extraction process. Dithiothreitol was added to the protein solution to achieve a final concentration of 5 mM and incubated at 56 °C for 30 min. After that, iodoacetamide was added to achieve a final concentration of 11 mM and incubated for 15 min at room temperature in the dark. Finally, the urea concentration of the sample was diluted to less than 2 M. Pancreatin was added at a mass ratio of 1:50 (pancreatin: protein) and incubated overnight at 37 °C. Then, pancreatin was added at a mass ratio of 1:100 (pancreatin: protein), and enzymatic hydrolysis was continued for 4 h. The peptides digested by trypsin were desalted with Strata X C18 (Phenomenex, Torrance County, NM, USA) and then freeze-dried under vacuum. The peptide was dissolved in 0.5 M TEAB and labeled following the TMT kit operating instructions: Thaw the labeling reagent, dissolve it in acetonitrile, mix it with the peptide, and incubate at room temperature for 2 h. Then, mix the labeled peptide, desalinate it, and freeze-dry it under vacuum. Utilize HPLC fractionation in conjunction with liquid chromatography-mass spectrometry. The peptides were fractionated by high-pH reverse-phase HPLC and then dissolved in mobile phase A of liquid chromatography (0.1% (*v*/*v*) formic acid aqueous solution) and separated using the EASY-nLC 1000 ultra-high performance liquid system. The obtained mass spectrum data were analyzed using MaxQuant (v1.5.2.8).

### 4.6. Differential Genes and Proteins GO Analysis

The analysis of differential genes and proteins was further accomplished through bioinformatics, including Gene Ontology (GO, https://geneontology.org, accessed on 1 December 2023) analysis, protein domain annotation, and Kyoto Encyclopedia of Genes and Genomes (KEGG, https://www.genome.jp/kegg, accessed on 1 December 2023) pathway annotation. Differential proteins with a variation greater than 1.5 were selected for further acetylation modification analysis.

### 4.7. Real Time Quantative PCR (qRT-PCR)

RNA extraction was the same as RNA-seq. Reverse transcription was performed using the HiScript III 1st Strand cDNA Synthesis Kit (+gDNA wiper) (Vazyme, Nanjing, China). Four primers were designed using Primer Premier 5. The RT-qPCR was performed by ChamQ Universal SYBR qPCR Master Mix (Vazyme). The reference gene used was UBQ, and it was analyzed using the 2^−ΔΔct^ method. Four technical replicates were carried out for each sample to ensure reproducibility and reliability. Statistical analysis of variance (ANOVA) was accomplished with SPSS Version 16.0. The significance level was set at *p* < 0.05.

## 5. Conclusions

Adding s-metolachlor can cause phytotoxicity in rice, and adding gibberellin can alleviate it. GA_3_ eliminates ROS by enhancing the activity of oxidoreductase, thus alleviating the phytotoxicity of s-metolachlor on rice plants. GA_3_ can influence the levels of flavonoids associated with oxidation-reduction processes and enhance the expression of related genes.

GA_3_ has the ability to alleviate the phytotoxicity of S-metolachlor to rice. Therefore, the dryland herbicide S-metolachlor could be utilized as a pre-emergence herbicide for weed control in rice fields when GA_3_ is added as a safener. This study has practical value for expanding the application range of S-metolachlor. It also holds significance for weed control and herbicide resistance management.

## Figures and Tables

**Figure 1 plants-13-02517-f001:**
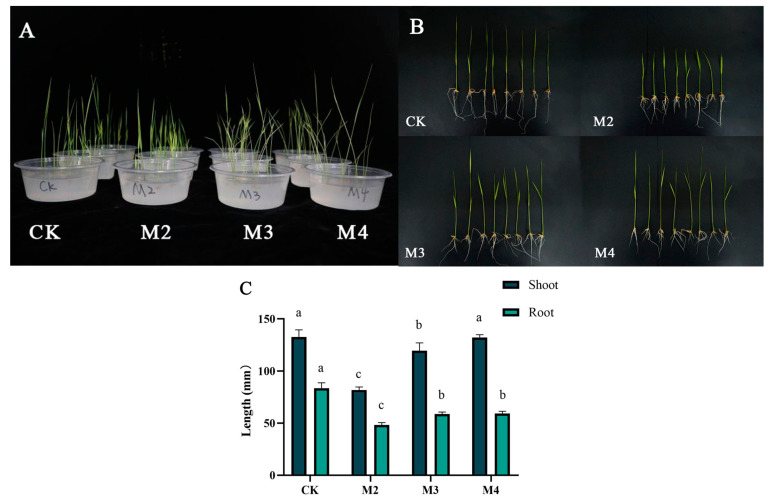
(**A**,**B**) The rice with different treatments after 7d. CK-M4 are different treatments to rice. (**C**) Shoot length, root length in different treatments. CK: blank control. M2: adding s-metolachlor. M3: adding s-metolachlor and GA_3_. M4: adding GA_3_. Standard error and significant difference are calculated by IBM SPSS Statistics 20.0. abc response 5% significant level.

**Figure 2 plants-13-02517-f002:**
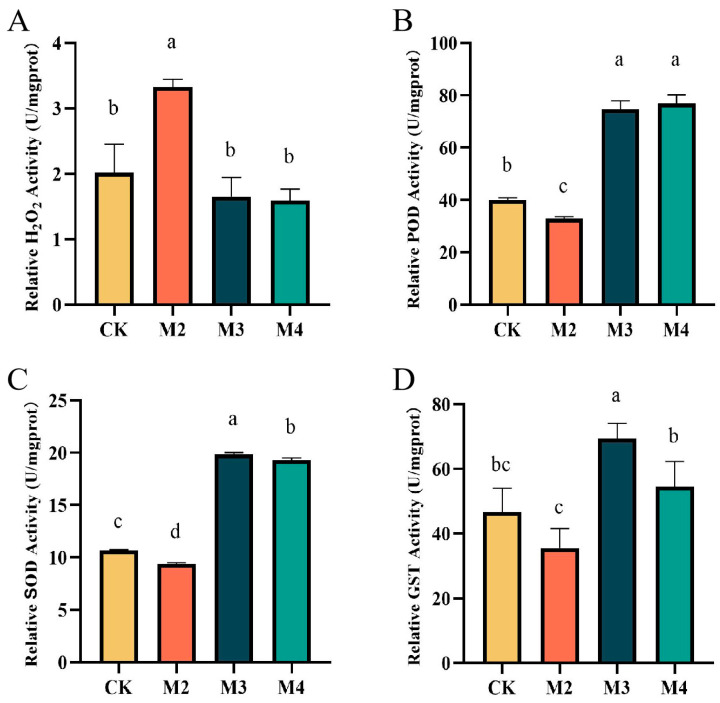
Enzymatic activity of rice. X-axis: Different Groups. Y-axis: (**A**) H_2_O_2_ content of rice. (**B**) relative peroxidase (POD) activity. (**C**) relative superoxidase dismutase (SOD) activity. (**D**) relative glutathione s-transferase (GST) activity. CK: Blank control. M2: Adding s-metolachlor. M3: Adding s-metolachlor and GA_3_. M4: Adding GA_3_. Standard error and significant difference are calculated by IBM SPSS Statistics 20.0. abcd response 5% significant.

**Figure 3 plants-13-02517-f003:**
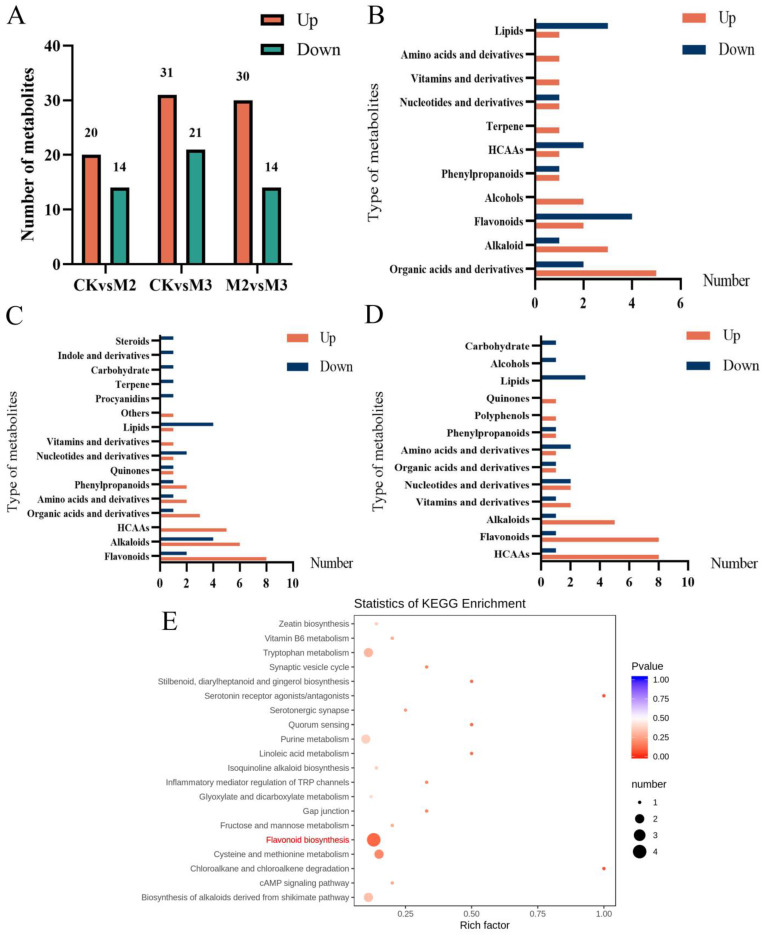
The metabolites of different treatments. (**A**) The number of different metabolites. (**B**–**D**) Metabolite numbers in different treatments. (**B**) CKvsM2 (The control group was CK) (**C**) CKvsM3 (The control group was CK) (**D**) M2vsM3 (The control group was M2). The horizontal coordinate represents the type of metabolite, and the vertical coordinate represents the amount of that type of metabolite in this group. (**E**) Bubble chart of KEGG pathway enrichment in M2vsM3.

**Figure 4 plants-13-02517-f004:**
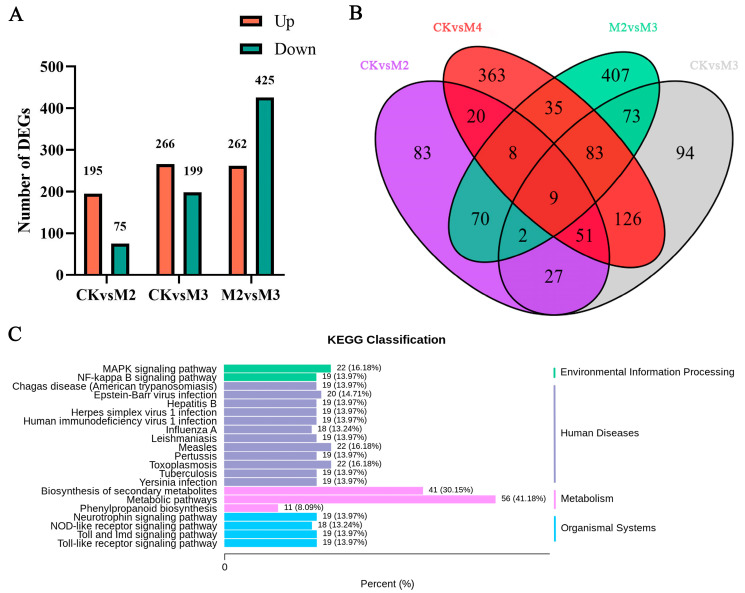
DEGs of M2vsM3. (**A**) The number of DEGs in different groups. (**B**) Venn diagram of DEGs (**C**) The number of DEGs in different pathways. DEGs are selected by annotated in pathway over 10. The control group was M2.

**Figure 5 plants-13-02517-f005:**
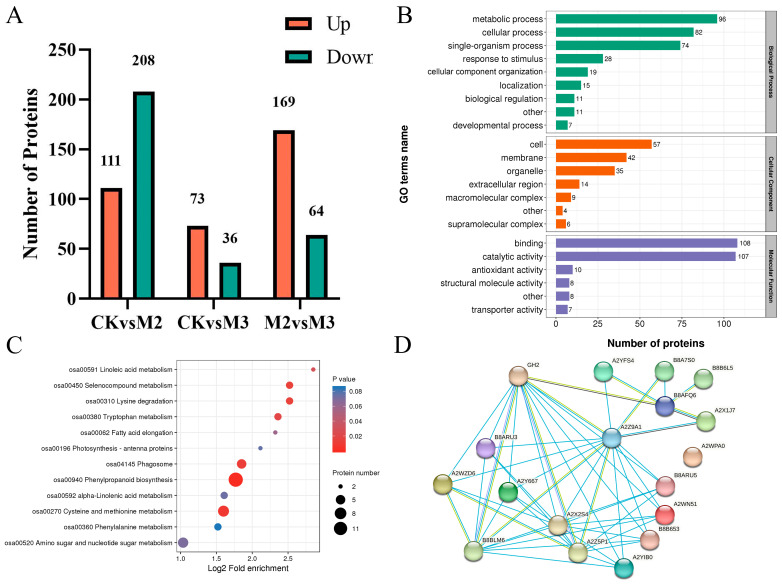
Histogram of differential proteins. (**A**) The number of different proteins. (**B**) GO function classify of proteins. (**C**) KEGG pathway enrichment of proteins. (**D**) Protein–protein interaction network. Red bubble stands for the core protein of this network. The control group was M2.

**Figure 6 plants-13-02517-f006:**
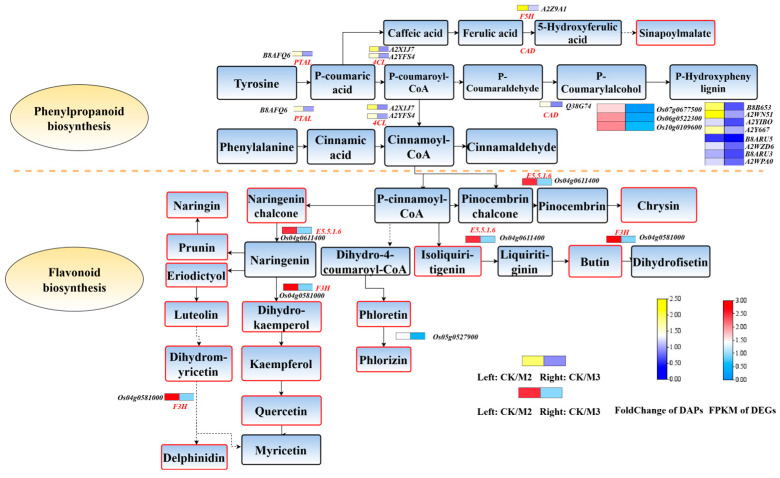
Phenylpropanoid and flavonoid biosynthesis pathway. The orange block shows differential metabolites in pathways. Red and blue indicate proteins, green and red indicate metabolites, respectively. *PTAL*, phenylalanine/tyrosine ammonia-lyase; *4CL*, 4-coumarate--CoA ligase; *CAD*, cinnamyl-alcohol dehydrogenase; *F5H*, ferulate-5-hydroxylase; *COMT*, caffeic acid 3-O-methyltransferase; *E1.11.1.7*, peroxidase; *F3H*, naringenin 3-dioxygenase; *E5.5.1.6*, chalcone isomerase; *PGT1*, phlorizin synthase.

**Figure 7 plants-13-02517-f007:**
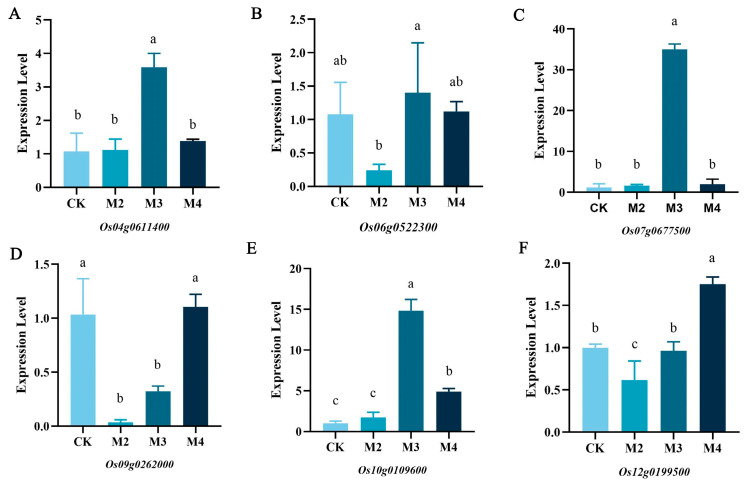
qRT-PCR of differential expression genes. (**A**) *Os04g061140*; (**B**) *Os06g0522300*; (**C**) *Os07g0677500*; (**D**) *Os09g0262000*; (**E**) *Os10g0109600*; (**F**) *Os12g0199500*. The horizontal axis represents the control group and the treatment group. The vertical axis represents the expression levels of differential genes. Statistical analysis of variance (ANOVA) with SPSS Version 16.0. The significance level was set at *p* < 0.05. abc response 5% significant.

**Table 1 plants-13-02517-t001:** DAPs in ko00940.

DAPs	CKvsM2		CKvsM3		M2vsM3		
	Fold Change	*p*-Value	Fold Change	*p*-Value	Fold Change	*p* Value	Protein Description
B8AFQ6	1.5456	6.1 × 10^−5^	0.9837	0.15610	1.5713	4.3 × 10^−5^	phenylalanine ammonia-lyase [*Oryza sativa Japonica* Group]
A2X1J7	1.9066	0.00044	0.9110	0.02346	2.0928	1.5 × 10^−6^	probable 4-coumarate–CoA ligase 3 [*Oryza sativa Japonica* Group]
A2YFS4	1.5471	0.00138	0.9791	0.56018	1.5801	0.00053	probable 4-coumarate–CoA ligase 4 [*Oryza sativa Japonica* Group]
Q38G74	1.4559	0.00286	0.8860	0.00106	1.6433	3.7 × 10^−5^	Cinnamyl-alcohol dehydrogenase *OS = Oryza sativa* subsp. *indica* OX = 39946 GN = GH2 PE = 3 SV = 1
**A2WN51**	2.3890	0.00012	0.9935	0.75944	2.4047	0.00012	Peroxidase *OS = Oryza sativa* subsp. *indica* OX = 39946 GN = OsI_01277 PE = 3 SV = 1
**A2WPA0**	1.2030	0.01086	0.7691	0.00020	1.5641	0.00026	peroxidase 1 [*Oryza sativa Japonica* Group]
**A2WZD6**	1.2390	0.00050	0.7602	2.1 × 10^−5^	1.6300	1.5 × 10^−5^	Peroxidase *OS = Oryza sativa* subsp. *indica* OX = 39946 GN = OsI_05314 PE = 3 SV = 1
**A2Y667**	1.7234	0.00010	1.0376	0.07656	1.6609	0.00012	Peroxidase *OS = Oryza sativa* subsp. *indica* OX = 39946 GN = OsI_20490 PE = 3 SV = 1
**B8ARU3**	1.0433	0.19318	0.6404	4.2 × 10^−6^	1.6290	9.6 × 10^−5^	Peroxidase *OS = Oryza sativa* subsp. *indica* OX = 39946 GN = OsI_18017 PE = 3 SV = 1
**B8ARU5**	0.5657	2.4 × 10^−5^	0.3445	3.6 × 10^−5^	1.6420	0.00072	Peroxidase *OS = Oryza sativa* subsp. *indica* OX = 39946 GN = OsI_18021 PE = 3 SV = 1
**B8B653**	1.9473	0.00090	0.6799	0.00024	2.8643	0.00016	Peroxidase *OS = Oryza sativa* subsp. *indica* OX = 39946 GN = OsI_27330 PE = 3 SV = 1
**A2YIB0**	/	/	1.542	0.00894	1.874	0.00084	Peroxidase OS = *Oryza sativa* subsp. *indica* OX = 39946 GN = OsI_24946 PE = 3 SV = 1
A2Z9A1	2.1725	0.00026	1.1701	0.05896	1.8567	0.00028	cytochrome P450 84A1 [*Oryza sativa Japonica* Group]

## Data Availability

All data are presented in the article.

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
