# Peer review of "Multiomics Analysis of the Mechanism by Which Gibberellin Alleviates S-Metolachlor Toxicity in Rice Seedlings"

_plants, 2024, doi:10.3390/plants13172517_

Round 1
Reviewer 1 Report
Comments and Suggestions for Authors
Some areas can be improved as seen in the peer review report.
Reviewer 2 Report
Comments and Suggestions for Authors
General comments
The study explores the phytotoxicity of s-metolachlor on rice seedlings, identifying GA3 as a safener to mitigate its adverse effects. It also reveals flavonoid metabolism and suggests combining GA3 with s-metolachlor for effective weed management. However, the study touches on transcriptome analysis but likely does not encompass every gene involved in the detoxification process. Conducting a more detailed analysis could provide a richer understanding of the genetic responses in rice seedlings to s-metolachlor and GA3 treatments.
Specific comments
1. Novelty: Line 77: The central question should be clearly defined at the end of the introduction section. To advance current knowledge, it is important to highlight what is novel about the research. The authors should revise the introduction to emphasize the novelty and hypothesis before outlining the aims of the study.
Overall, the manuscript includes only six recent citations from 2020. It would be beneficial to incorporate more recent references, especially in the Discussion section.
2. Scope: The work fits the Plants.
3. Quality: The data and analysis presentation was appropriate. However, I believe the authors should improve Figure 2A. For example, in lines 91 to 93, the authors describe that "The addition of exogenous GA3 significantly alleviated the phytotoxicity (Fig. 1A), and the mitigation effect was mainly observed in the stem and leaf parts of the crop." While Figure 2B effectively shows the results, the photograph in Figure 2A does not visualize the mitigation of the effects. The authors should consider enhancing the photograph (perhaps by adjusting the lighting) and increasing its size for better visualization of the effects.
Figure 7 could also be colored like the others
Which test was applied after the comparative analysis and post-hoc ANOVA? It is important to specify the test used in the figure captions, along with the p-values.
4. Scientific soundness: The methodology is thorough and well-described, indicating that the study was conducted effectively.
5. Interest to the Readers: This manuscript will be of great interest to scientists in the field. The findings offer practical solutions for improving agricultural practices and understanding herbicide interactions with plant physiology.
6. Overall Merit: It is a good work. The conclusions provide valuable insights into the interaction between GA3 and s-metolachlor, offering a pathway for improving weed management strategies in rice cultivation.
7. English Level: The authors and editorial staff should carefully revise the work. There are some hard-to-hard sentences and grammatical mistakes.
Attention:
Replace the current keywords "Rice; S-metolachlor; Gibberellin”, which are described in the title of the manuscript.
Line 23: “the elimination of reactive oxygen species (ROS) in plants.”
Line 27: “peroxidase (POD) activity”
Lines 27-28: change “reactive oxygen species (ROS)” by “ROS”
Lines 32 to 36: This paragraph is out of context. It should be rewritten, shortened, and added to line 31 as a strong conclusion.
Lines 56 to 58: This paragraph is off-topic. The authors describe information about metolachlor, but the reference by Liu et al. (2019) discusses the role of glyphosate in algae, with no apparent connection between the subjects.
Line 224: Metolachlor is not a medicine!
Line 382: “eliminates ROS by...”
Line 380: The conclusion should be moved to line 305, before the Material and Methods section.
Comments on the Quality of English LanguageA careful proofreading is necessary.
Round 2
Reviewer 2 Report
Comments and Suggestions for Authors
The authors included all my suggestions